# Structural definition of a neutralization epitope on the N-terminal domain of MERS-CoV spike glycoprotein

Haixia Zhou [1,6], Yingzhu Chen[2,3,6], Shuyuan Zhang[1], Peihua Niu[2], Kun Qin[2], Wenxu Jia[4], Baoying Huang[2], Senyan Zhang[1], Jun Lan[1], Linqi Zhang[4], Wenjie Tan[2] & Xinquan Wang[1,5]

Most neutralizing antibodies against Middle East respiratory syndrome coronavirus (MERS-CoV) target the receptor-binding domain (RBD) of the spike glycoprotein and block its binding to the cellular receptor dipeptidyl peptidase 4 (DPP4). The epitopes and mechanisms of mAbs targeting non-RBD regions have not been well characterized yet. Here we report the monoclonal antibody 7D10 that binds to the N-terminal domain (NTD) of the spike glyco-protein and inhibits the cell entry of MERS-CoV with high potency. Structure determination and mutagenesis experiments reveal the epitope and critical residues on the NTD for 7D10 binding and neutralization. Further experiments indicate that the neutralization by 7D10 is not solely dependent on the inhibition of DPP4 binding, but also acts after viral cell attachment, inhibiting the pre-fusion to post-fusion conformational change of the spike. These properties give 7D10 a wide neutralization breadth and help explain its synergistic effects with several RBD-targeting antibodies.

[1] The Ministry of Education Key Laboratory of Protein Science, Beijing Advanced Innovation Center for Structural Biology, Beijing Frontier Research Center for Biological Structure, Collaborative Innovation Center for Biotherapy, School of Life Sciences, Tsinghua University, 100084 Beijing, China. [2] Key Laboratory of Medical Virology, National Health and Family Planning Commission, National Institute for Viral Disease Control and Prevention, China CDC, 102206 Beijing, China. [3] Key Laboratory of Carcinogenesis and Translational Research (Ministry of Education), Department of Clinical Laboratory, Peking University Cancer Hospital & Institute, 100142 Beijing, China. [4] Comprehensive AIDS Research Center, Collaborative Innovation Center for Diagnosis and Treatment of Infectious Diseases, Department of Basic Medical Sciences, School of Medicine,  Tsinghua University, 100084 Beijing, China. [5] Collaborative Innovation Center for Biotherapy, State Key Laboratory of Biotherapy and Cancer Center, West China Hospital, West China Medical School, Sichuan University, 610065 Chengdu, China. [6] These authors contributed equally: Haixia Zhou, Yingzhu Chen. Correspondence and requests for materials should be addressed to W.T. (email: tanwj28@163.com) or to X.W. (email: xinquanwang@mail.tsinghua.edu.cn)

Middle East respiratory syndrome coronavirus (MERS-CoV), a novel lethal human virus in the family of Coronaviridae, was first identified in Saudi Arabia in June 2012[1]. Infection by this pathogen causes an acute respiratory disease designated as MERS, with symptoms that are very similar to those of SARS[2]. Globally, MERS-CoV infections have been confirmed in 27 countries causing 803 deaths (http://www.who.int/emergencies/mers-cov/en/). Interspecies transmission from dromedary camels to humans is considered to be one major route of transmission in the Middle East region[3,4]. However, many infected patients without camel exposure and a recent MERS outbreak in Korea demonstrated that large-scale human-to-human transmissions can occur through close contacts[5]. Due to its potential for mutating toward efficient human-to-human transmission and causing a pandemic, MERS-CoV was listed as a Category C Priority Pathogen by the US National Institute of Allergy and Infectious Diseases.

Monoclonal antibodies (mAbs) with potent neutralizing activity have become promising candidates for both prophylactic and therapeutic interventions against viral infections[6]. On coronaviruses, the component primarily targeted by mAbs is the homotrimeric spike (S) glycoprotein of the virion. As a typical class I fusion glycoprotein, the S trimer of highly pathogenic coronaviruses such as MERS-CoV and SARS-CoV, which mediates receptor recognition and membrane fusion during viral entry[7–12], undergoes protease cleavage into the S1 and S2 subunits, positional change of the receptor-binding domain (RBD) in the S1 subunit for receptor binding, dissociation of the S1-receptor complex, and finally formation of a six-helix bundle by the S2 subunits. A series of RBD-targeting antibodies against MERS-CoV, which block the binding of the S trimer to the cellular receptor DPP4, have been reported and characterized[13–22]. These antibodies exhibited high potency in inhibiting the infectivity of pseudotyped and live MERS-CoV in cells and animal models. The neutralizing epitopes and mechanisms of antibodies including 4C2, D12, m336, MERS-27, JC57-14, CDC-C2, MERS-4, and MERS-GD27 were further elucidated at the atomic level by structural and functional studies[19–25].

Sequence comparisons of different MERS-CoV strains have shown that most naturally occurring mutations of the S glycoprotein are located on the RBD of the S1 subunit and the S2 subunit. Considering the rapid evolution and high genome variation of RNA viruses, more mutations on the RBD may enable the new strains to escape neutralization by currently known RBD-targeting antibodies. Therefore, new mAbs targeting other functional regions of the MERS-CoV S glycoprotein and/or neutralizing by different mechanisms are important for developing effective prophylactic and therapeutic interventions against MERS-CoV infection.

Although several mAbs targeting non-RBD regions have recently been reported, their neutralizing epitopes and mechanisms remain unclear[20,21,26]. In this study, we isolated and characterized the mouse mAb 7D10 by combining structural, biochemical, and functional studies. The 7D10 antibody recognizes the NTD of MERS-CoV S glycoprotein and neutralizes the infectivity of pseudotyped and live virus with a potency comparable to those of the most active RBD-targeting antibodies. We also found that the epitope and mechanism of 7D10, which are different from those of RBD-targeting antibodies, enable it to have a better neutralizing breadth and to work synergistically with other antibodies against different MERS-CoV strains. All these results indicate that 7D10 is a very promising candidate for the future combined use of different antibodies in our battle against MERS-CoV.

## Results

**Characterization of neutralizing mAb 7D10 targeting the NTD.** To generate MERS-CoV neutralizing mAbs with epitopes outside the RBD, mice were immunized with recombinant MERS-CoV S protein (residues 1–1297). Subsequently, the spleenocytes were harvested and fused with SP2/0 myeloma cells, and the hybridoma cell lines were screened for positive clones by ELISA with the S protein[27]. The positive clones were further tested for their reactivity to different S fragments, including the S1 subunit NTD (residues 18–353), RBD (residues 367–606), and the S2 subunit (residues 726–1297). One NTD-specific mAb, named as 7D10, was finally isolated with an $EC_{50}$ of approximately 0.31 µg mL$^{-1}$ in ELISA (Fig. 1a). It exhibited no cross-reactivity with the RBD at a concentration of 4 µg mL$^{-1}$ (Fig. 1b). We further assessed the potential of 7D10, in the form of crude extracts from mouse ascites, for inhibiting MERS-CoV entry into susceptible Huh7 cells and Vero E6 cells with either pseudotyped or infectious viruses. As expected, 7D10 was able to neutralize the infectivity of pseudotyped and live MERS-CoV (Fig. 1c, d). The neutralizing activity of 7D10 was dose-dependent, with an $IC_{50}$ of approximately 0.18 µg mL$^{-1}$ against pseudotyped virus and practically the same $IC_{50}$ of approximately 0.2 µg mL$^{-1}$ against live virus (EMC strain) (Fig. 1c, d). Images illustrating the reduced PFU formation, corresponding to the rate of neutralization of live MERS-CoV, are shown in Fig. 1e. Antibody isotyping showed that 7D10 belongs to the IgG1 subtype. Sequencing further determined that the heavy chain germline V and J segments are IGHV1-12*01 and IGHJ2*03, while those of light chain are IGKV03-12*01, IGKJ1*01, and IGKJ1*02, respectively (Supplementary Table 1).

We also generated a chimeric version of 7D10 (7D10-H) by combining the V segments of 7D10 with the human IgG1 backbone, which was efficiently expressed and purified in FreeStyle 293-F cells (Supplementary Fig. 1A). The bio-layer interferometry (BLI) experiment showed that the affinity constant of the binding between 7D10-H and NTD was approximately 25 nM (Table 1 and Supplementary Fig. 1B). The $IC_{50}$ of the purified 7D10-H against cell entry by pseudotyped MERS-CoV was approximately 0.06 µg mL$^{-1}$ (Supplementary Fig. 1C). We also investigated the protective efficacy of 7D10-H against infection of pseudotyped MERS-CoV using R26-hDPP4 mice model with a human DPP4 inserted into the Rosa26 locus by CRISPR/Cas9, which could also been productively infected by high-titer MERS-CoV pseudovirus, with effects comparable to the authentic infection[28]. Bioluminescence of the Fluc reporter showed that the pseudovirus infection in the mice was clearly prevented by 7D10-H and RBD-specific mAb MERS-4 when both antibodies were administered by the intraperitoneal injection with a dose of 200 µg per mouse (Supplementary Fig. 1D). The recombinant chimeric 7D10-H, which retained the activities as the mouse 7D10 and protected R26-hDPP4 mice against challenge of pseudotyped MERS-CoV, was utilized in subsequent binding and neutralization experiments.

**Overall structure of the 7D10 scFv bound to the NTD.** To structurally characterize the 7D10 and its binding to the spike protein, we determined the crystal structure of the antibody scFv (7D10-scFv) in complex with the NTD at a resolution of 3.0 Å with a final $R_{work}$ of 0.188 and $R_{free}$ of 0.224. Statistics of diffraction data collection, processing, and structure refinement are listed in Table 2. There were three complexes of 7D10-scFv bound to NTD per asymmetric unit. The refined model contains residues Tyr18 to Ser353 of MERS-CoV NTD, Glu1 to Ser120 of the $V_H$ and Asp26 to Lys136 of the $V_L$. N-linked glycans attached to Asn66, Asn104, Asn125, Asn155, Asn166, Asn222, Asn236,

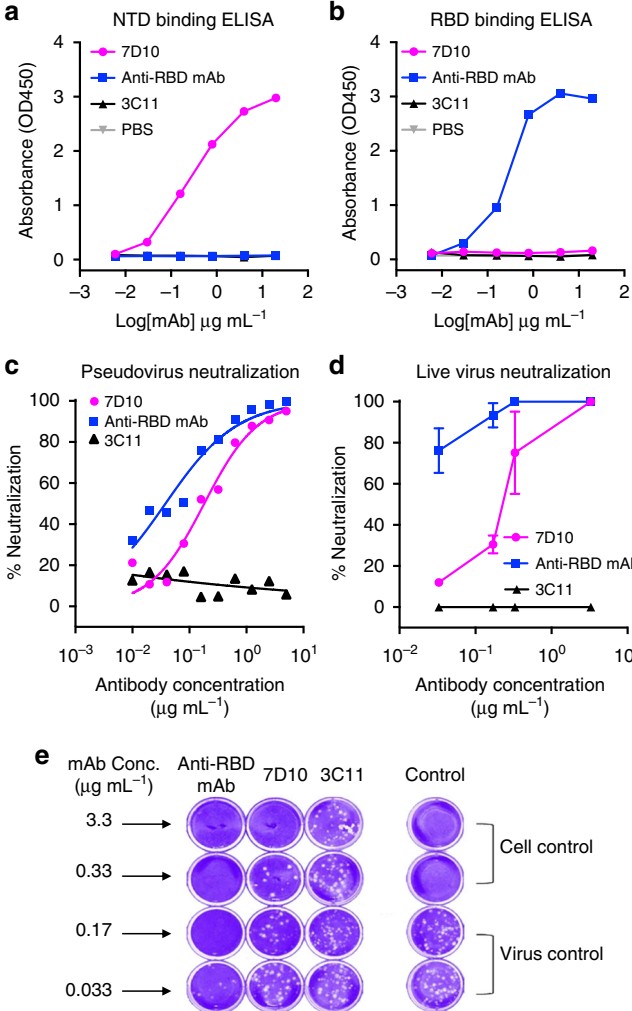

**Table 1 7D10-H binding affinities with and neutralization of mutant NTDs or pesudoviruses**

| Analyte | kD (M) | Fold decrease | IC$_{50}$ (µg mL$^{-1}$) | Fold decrease |
|---|---|---|---|---|
| WT | 2.50E−08 | 1.00 | 0.057 | 1 |
| Y18A | 1.73E−07 | 6.93 | 0.583 | 10.16 |
| D20A | 2.14E−07 | 8.57 | 3.454 | 60.17 |
| P23A | 2.57E−07 | 10.29 | 0.368 | 6.42 |
| D24A | N.D. | – | N.D. | – |
| V26A | 3.05E−07 | 12.20 | 10.440 | 181.88 |
| S28A | 1.31E−07 | 5.24 | 0.221 | 3.86 |
| E188A | 3.70E−06 | 148.00 | N.D. | – |
| S191A | 5.95E−08 | 2.38 | 0.210 | 3.65 |
| N222Q | 2.80E−06 | 112.00 | 2.869 | 49.98 |
| N226A | 1.84E−07 | 7.35 | 0.765 | 13.32 |
| L234A | 1.67E−07 | 6.70 | 0.883 | 15.38 |
| R235A | N.D. | – | N.D. | – |
| N236A | 1.77E−07 | 7.08 | 0.062 | 1.08 |

*N.D.* no binding or no pseudovirus neutralization observed

**Fig. 1** 7D10 binding specificity and neutralization potency. Recombinant NTD or RBD of MERS-CoV S glycoprotein at 1 µg mL$^{-1}$ were used to coat plates overnight at 4 °C, and each of the mAbs including 7D10, an antibody anti-RBD (MERS-GD27) and an unrelated antibody (3C11) were serially diluted in PBS and assessed for binding affinity and specificity to the NTD (**a**) and RBD (**b**). **c** Neutralization of the pseudotyped MERS-CoV. DPP4-expressing Huh7 cells were cultured with 200 TCID$_{50}$ of pseudotyped MERS-CoV in the presence of serially diluted mAbs. The neutralization percentage was calculated by measuring luciferase expression compared to the pseudotyped virus-infected cell control. **d** Neutralization of live MERS-CoV. Different concentrations of the mAbs were pre-cultured with the live MERS-CoV (EMC strain) in Vero E6 cell monolayers. The neutralization percentage was evaluated by calculating the decrease in plaque number compared with the virus-infected control. Data are shown as mean ± SD. **e** PFU images of viral infection in the presence of the mAbs on day 3. The images correspond to the neutralizing percentages in (**d**). Approximately, 30–35 PFU virus stocks (EMC strain) were used to infect Vero E6 cells in a 12-well plate with or without mAbs. MERS-GD27, 3C11 mAbs, and PBS were used as the positive, unrelated, and blank controls, respectively. Source data are provided as a Source Data file

and Asn244 of the NTD are also included in the model. It has been previously shown that the MERS-CoV NTD folds into a galectin-like structure, which can be separated into top, core and bottom subdomains (Fig. 2a). Upon binding, the 7D10-scFv contacts the top subdomain of the NTD and the Asn222-linked glycans with its heavy and light chains (Fig. 2a and Supplementary Fig. 2). All three CDRs of the heavy chain and the CDR1 and CDR3 of the light chain participate in the binding (Fig. 2a). The

**Table 2 Data collection and refinement statistics**

| | 7D10-scFv with the NTD |
|---|---|
| **Data collection** | |
| Space group | $P2_12_12_1$ |
| Cell dimensions | |
| $a, b, c$ (Å) | 107.57, 180.36, 245.16 |
| $\alpha, \beta, \gamma$ (°) | 90, 90,90 |
| Resolution (Å) | 50-3.0 (3.07-3.0) |
| $R_{merge}$ | 0.178 (1.231) |
| $R_{pim}$ | 0.050 (0.344) |
| CC$_{1/2}$ | 0.832 |
| I / σI | 21.2 (3.3) |
| Completeness (%) | 99.9 (100) |
| Redundancy | 13.3 (13.7) |
| **Refinement** | |
| Resolution (Å) | 44.93-3.0 |
| No. reflections | 95996 |
| $R_{work}$ / $R_{free}$ (%) | 18.76/22.36 |
| No. atoms | |
| Protein | 13347 |
| Ligand (Glycan) | 788 |
| $B$-factors (Å$^2$) | |
| Protein | 64.44 |
| Ligand (Glycan) | 95.81 |
| R.m.s. deviations | |
| Bond lengths (Å) | 0.012 |
| Bond angles (°) | 1.41 |
| Ramachandran plot (%) | |
| Most favored | 94 |
| Allowed | 5.3 |
| Disallowed | 0.3 |

*Values in parentheses are for highest-resolution shell. One crystal was collected for structure

buried surface between the 7D10-scFv and the NTD encompasses approximately 551 Å$^2$ for the heavy chain and 320 Å$^2$ for the light chain.

**Structural features of the interface between 7D10 and NTD.** The binding interface between 7D10-scFv and NTD consists of 12 residues and Asn222-linked glycans from the NTD, as well as 15 residues from all 6 CDRs except for LCDR2 (Fig. 2b, c). The interacting residues from the NTD are Tyr18, Asp20, Pro23, Asp24, Val26, Ser28, Glu188, Ser191, Asn226, Lue234, Arg235,

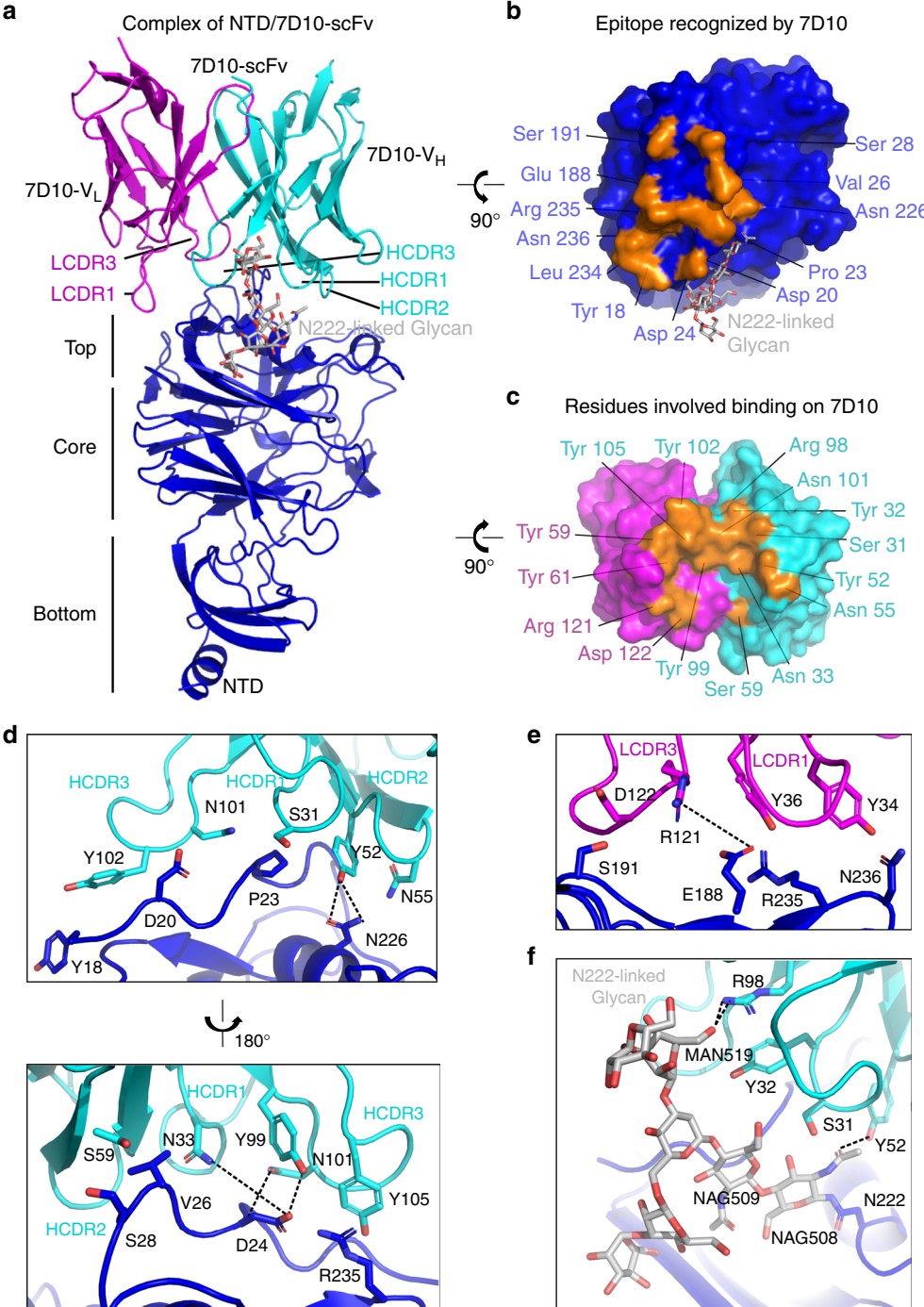

**Fig. 2** Crystal structure of 7D10-scFv bound to NTD and the binding interface. **a** An overall structure of the NTD/7D10-scFv complex in which the NTD, N222-linked glycans on the NTD, 7D10 V$_L$, and 7D10 V$_H$ are colored in blue, gray, magenta, and cyan, respectively. **b** Epitope on the NTD recognized by 7D10. The NTD is represented as blue surface, on which the protein region bound by 7D10 is displayed in orange and the N222-linked glycans are displayed as gray sticks. **c** 7D10 residues that are involved in the binding. The V$_L$ and V$_H$ are colored in magenta and cyan, respectively, and the residues interacting with 7D10 are displayed in orange. **d** Interactions between the 7D10 V$_H$ residues and the corresponding residues of NTD. **e** Interactions between the 7D10 V$_L$ residues and the corresponding residues of NTD. **f** Zoom-in view of interactions between N222-linked glycans and 7D10

and Asn236. Together with the Asn22-linked NAG508, NAG509 and MAN519, they form the conformational epitope recognized by 7D10 (Fig. 2b). The residues recognizing 7D10 are Ser31, Tyr32, Asn33 from the HCDR1, Tyr52, Asn55, and Ser59 from the HCDR2, Arg98, Tyr99, Asn101, Tyr102, and Tyr105 from the HCDR3, Tyr59, and Tyr61 from the LCDR1, and Arg121 and Asp122 from the LCDR3 (Fig. 2c). Specifically, 7D10 HCDR1 residues Ser31, Tyr32, and Asn33 interact with Pro23 and Asp24

from the NTD, and a formed hydrogen bond is from 7D10 Asn33 to NTD Asp24 (Fig. 2d and Supplementary Table 2). The 7D10 HCDR2 contributes to the recognition via Tyr52, Asn55, and Ser59 to interact Val26, Ser28, and Asn226 of the NTD, and two hydrogen bonds are formed between 7D10 Tyr52 and NTD Asn226 (Fig. 2d and Supplementary Table 2). Compared with HCDR1 and HCDR2, the HCDR3 engages the NTD more extensively with its Arg98, Tyr99, Asn101, Tyr102, and Y105

interacting with Tyr18, Asp20, Pro23, Asp24, and Arg235 of the NTD (Fig. 2d). Tyr99 and Asn101 of 7D10 form two hydrogen-bonding interactions with Asp24 of the NTD (Supplementary Table 2). For the light chain, the LCDR1 and LCDR3 residues Tyr59, Tyr61, Arg121, and Asp122 interact with Glu188, Ser191, Arg235, and Asn236 of the NTD, and a salt bridge is formed between Arg121 of LCDR3 and Glu188 of the NTD (Fig. 2e and Supplementary Table 2). A prominent feature at the interface is the extensive recognition of Asn222-linked glycans by all three heavy chain CDRs (Fig. 2f and Supplementary Table 3). Specific hydrogen-bonding interactions occur between Tyr58 and Arg98 of 7D10 and the NAG508 and MAN519 glycans, respectively (Fig. 2f and Supplementary Table 2).

**Confirmation of the neutralizing epitope.** To confirm the epitope and its critical residues, we performed a mutagenesis study by introducing single mutations to all 13 NTD recognized residues including Trp18, Asp20, Pro23, Asp24, Val26, Ser28, Glu188, Ser191, Asn222, Asn226, Lue234, Arg235, and Asn236. We first examined the effects of these NTD mutations on the binding by 7D10-H. The 7D10-H bound the wild-type NTD with an affinity of approximately 25 nM (Table 1 and Supplementary Fig. 3). By contrast, the D24A and R235A mutations dramatically reduced the binding, to a level that was undetectable by BLI experiment (Table 1 and Supplementary Fig. 2). The E188A and N222Q mutations reduced by the binding affinity by 148-fold to 3.7 μM and 112-fold to 2.8 μM, respectively (Table 1 and Supplementary Fig. 3). All the other nine mutations had variant unequal effects on the binding by reducing the affinity in the range of 2- to 15-fold (Table 1 and Supplementary Fig. 2). The effects of these mutations on the neutralizing activity of 7D10-H were in consistent with the changes of binding affinity. Pseudotyped MERS-CoV bearing D24A, E188A, or R235A mutation in the spike glycoprotein escaped the neutralization by 7D10-H (Table 1 and Supplementary Fig. 4). The $IC_{50}$ values of 7D10 against pseudotyped MERS-CoV bearing D20A, V26A, or N222Q mutation were increased approximately by 60-, 181-, and 50-fold (Table 1 and Supplementary Fig. 4). The binding and neutralization assays collectively revealed that Asp24, Val26, Glu188, Arg235, and Asn222-linked glycans are critical for recognition and neutralization of MERS-CoV by 7D10.

**7D10-H against pseudotyped MERS-CoV with natural mutations.** Sequencing of multiple clinical isolates had revealed that the MERS-CoV S glycoprotein is evolving at an average rate of $1.12 \times 10^{-3}$ substitutions per site per year[29]. Alignments of the deposited sequences in the NCBI identified 22 naturally changing residues from the prototype EMC sequence including V26F, V26I, V26A, D158Y, L411F, T424I, A482Y, L506F, D509G, V530L, V534A, E536K, D537E, V810I, Q833R, Q914H, R1020H, R1020Q, A1193S, T1202I, G1224S, and V1314A, which are located in the NTD (residues 18–353) and RBD (residues 367–606) of the S1 subunit, and the S2 subunit (residues 752–1297). Several residue changes on the RBD, such as those occurring on D506, D509, and E536, indeed enabled the MERS-CoV to escape the neutralization of antibodies targeting the RBD[20,21]. Considering that most of the mutations are outside the NTD, we speculated that 7D10-H would have a better tolerance for these naturally occurring mutations. We generated pseudotyped MERS-CoV bearing the EMC strain S glycoproteins and its mutants harboring all the 22 listed residue changes. The neutralization assays showed that 7D10-H showed effective neutralizing activity against almost all pseudotyped MERS-CoV variants. Only the two mutations V26F and V26A on the NTD increased the $IC_{50}$ value of 7D10-H by more than 150-fold and significantly reduced its neutralization activity (Fig. 3a, b), which

confirmed the results of the structural and biochemical studies of the binding interface. All other naturally occurring mutations, most of them on the RBD and the S2 subunit did not affect the neutralization capability of 7D10-H (Fig. 3a, b), indicating that 7D10 would have a wide neutralization breadth against different variants of MERS-CoV.

**Combination of 7D10 with other RBD-targeting antibodies.** The current available MERS-CoV antibody epitopes with solved structures are all on the RBD, which can be grouped into three categories: (1) epitope of MERS-4; (2) epitopes of MERS-27, D12, 4C2, and JC57-14; and (3) epitopes of m336, MCA1, CDC-C2, and the newly reported MERS-GD27 (Supplementary Fig. 5)[25]. In our study of the RBD-specific mAb MERS-4, we also found synergism with the NTD-targeting mAb 5F9[25]. Thus, the elucidation of the epitope targeted by 7D10, which added a category outside the RBD (Supplementary Fig. 5), prompted us to study the combined effect of 7D10 together with the three representative antibodies MERS-4, MERS-27, and MERS-GD27 in the neutralization of pseudotyped MERS-CoV by titrating the neutralizing potency of an equimolar mixture of the two antibodies and comparing the dose response with that observed in neutralization assays performed with the individual antibody alone. As shown in the Fig. 4, the combination index (CI) values of MERS-GD27 combined with 7D10 at FA values of effective dose 50%, 75%, 90%, and 95% ($ED_{50}$, $ED_{75}$, $ED_{90}$, and $ED_{95}$, respectively) were 0.26, 0.25, 0.24, and 0.24, respectively. As a CI value of 1 indicates an additive effect, <1 indicates synergism, and >1 indicates antagonism, the combination of 7D10 and MERS-GD27 worked in a clearly synergistic manner. Meanwhile, the combination index (CI) values of combined MERS-4 with 7D10 at FA values of effective dose 50%, 75%, 90%, and 95% ($ED_{50}$, $ED_{75}$, $ED_{90}$, and $ED_{95}$) were 0.25, 0.27, 0.30, and 0.33, respectively. Thus, the combination of MERS-4 and 7D10 also demonstrated synergism, in particular at relatively lower concentrations. However, the percent neutralization obtained using combined MERS-27 and 7D10 showed no obvious difference of half maximal inhibitory concentration ($IC_{50}$) compared with that of 7D10 alone. The combination index (CI) values of combined MERS-27 and 7D10 at FA values of effective dose 50%, 75%, 90%, and 95% ($ED_{50}$, $ED_{75}$, $ED_{90}$, and $ED_{95}$) were 0.82, 0.87, 0.94, and 1.00, respectively. It indicated that the combination of 7D10 with MERS-27 exhibited neither synergy nor antagonism.

**Mechanism of 7D10 neutralization.** A major reported MERS-CoV neutralization mechanism relies on inhibiting the binding of the S trimer with the cellular receptor DPP4. The epitopes of these reported antibodies all reside in the RBD responsible for receptor binding. The fact that the 7D10 epitope is outside the RBD indicated that it may have a different neutralizing mechanism. We first examined if 7D10 is still able to inhibit the receptor binding by the S trimer. The FACS analysis of cell-surface staining showed that the scFv and Fab fragments of 7D10-H did not inhibit the staining of Huh7 cells by the S trimer, while the 7D10-H slightly reduced the staining (Fig. 5a, Supplementary Table 4 and Supplementary Fig. 6). By contrast, the RBD-targeting mAb MERS-4 was much more potent than 7D10-H in inhibiting the binding of the S trimer to Huh7 cells. Moreover, the Fab and scFv fragments of MERS-4 retained nearly the same potency in the inhibition (Fig. 5a and Supplementary Table 4). Surface plasmon resonance (SPR) analysis confirmed these conclusions by showing that 7D10-H, and not its Fab or scFv fragments, could interfere with the binding of the S trimer to chip-coupled DPP4 in a dose-dependent manner (Supplementary

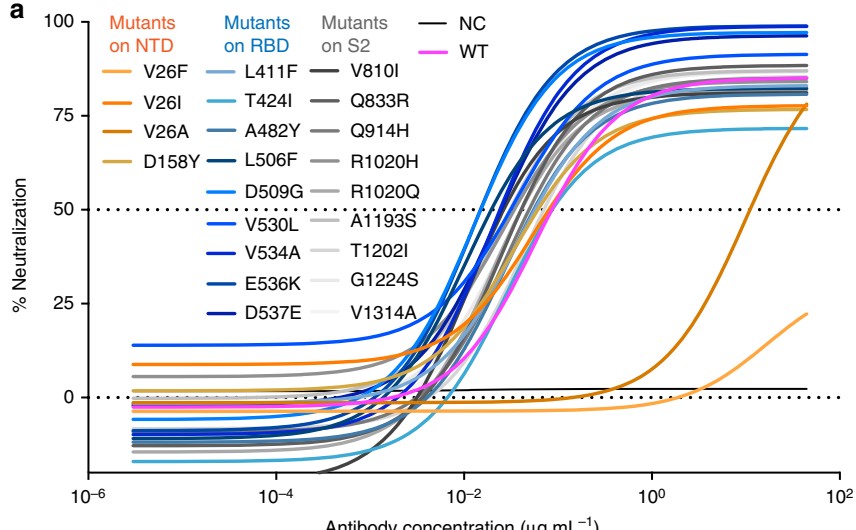

**Fig. 3** Breadth of 7D10-H neutralization. **a** Neutralizing analysis of 7D10-H against MERS-CoV wild-type (WT) and its variant mutants; site-directed mutations were introduced into the EMC strain to create 22 variant mutants according to natural mutations of MERS-CoV S. **b** Summary of 7D10-H mediated inhibition of infection by all pseudotyped viruses. $IC_{50}$ neutralization titers for mutant EMC S variants are presented relative to wild-type S (set to 1). Source data are provided as a Source Data file

Fig. 7), while the IgG, Fab, and scFv of MERS-4 all inhibited the binding (Supplementary Fig. 7).

To investigate why the IgG, Fab, and scFv of 7D10 inhibit receptor binding differently, we constructed models of their binding to the S trimer. The MERS-CoV S trimer structure was determined by cryo-EM with the RBD in standing or lying positions, and only the standing RBD could bind to the DPP4 receptor. After superimposing the NTD/7D10-scFv crystal structure onto the S trimer, we observed no steric clashes between three NTD-bound scFv fragments and one or two RBD-bound DPP4 receptors (Fig. 5b and Supplementary Fig. 8). The S trimer with three RBD-bound receptors was not considered because the cryo-EM study of the MERS-CoV S trimer only revealed conformations with one or two standing RBDs. When the scFv was replaced with the Fab, there were also no steric clashes between the Fab and DPP4 receptor (Fig. 5c). It is more complicated to model the binding of 7D10-H to the S trimer, considering that the IgG form has two binding sites and the intrinsic flexibility. We found that binding of the 7D10-H IgG to the NTD in certain orientations could inhibit the binding of DPP4 due to steric clashes, while there were still no steric clashes with the 7D10-H bound in some other orientations (Fig. 5d, e). These results provided a structural explanation for the inability of 7D10-H scFv and Fab to inhibit the binding of the S trimer to the DPP4 receptor. They may also explain why the 7D10-H IgG form

is not as potent as the MERS-4 IgG, Fab, and scFv which all directly bind to the RBD.

In parallel with biochemical studies, we also examined the neutralizing activities of 7D10-H IgG, Fab, and scFv. The 7D10-H Fab and scFv did not interfere with the binding of the S trimer to the DPP4 receptor. However, they were still able to inhibit the cell entry of pseudotyped MERS-CoV with $IC_{50}$ value of 0.26 µg mL$^{-1}$ and 0.28 µg mL$^{-1}$, respectively (Fig. 6a). Although the 7D10-H Fab and scFv are less active than the IgG in infection inhibition, they were still comparable to the Fab or scFv fragments of several reported RBD-targeting antibodies such as MERS-4 Fab ($IC_{50}$: 1.49 µg mL$^{-1}$) and MERS-4 scFv ($IC_{50}$: 0.55 µg mL$^{-1}$) (Supplementary Fig. 9A). These results collectively indicated that neutralization by 7D10-H involves other mechanism besides interfering with the initial receptor binding. We tested and compared the neutralizing activity of 7D10 in pre-attachment and post-attachment settings. After the cell attachment, 7D10 was still able to inhibit infection by pseudotyped MERS-CoV with an $IC_{50}$ of 0.55 µg mL$^{-1}$ (Fig. 6b). In comparison, MERS-4, which is more potent than 7D10 in inhibiting receptor binding, exhibited very weak neutralization after receptor binding (Supplementary Fig. 9B).

The above results, especially the retaining activity of 7D10 after viral attachment indicated that 7D10 would also interfere with the prefusion to postfusion conformational transition of the S glycoprotein required for membrane fusion. This transition

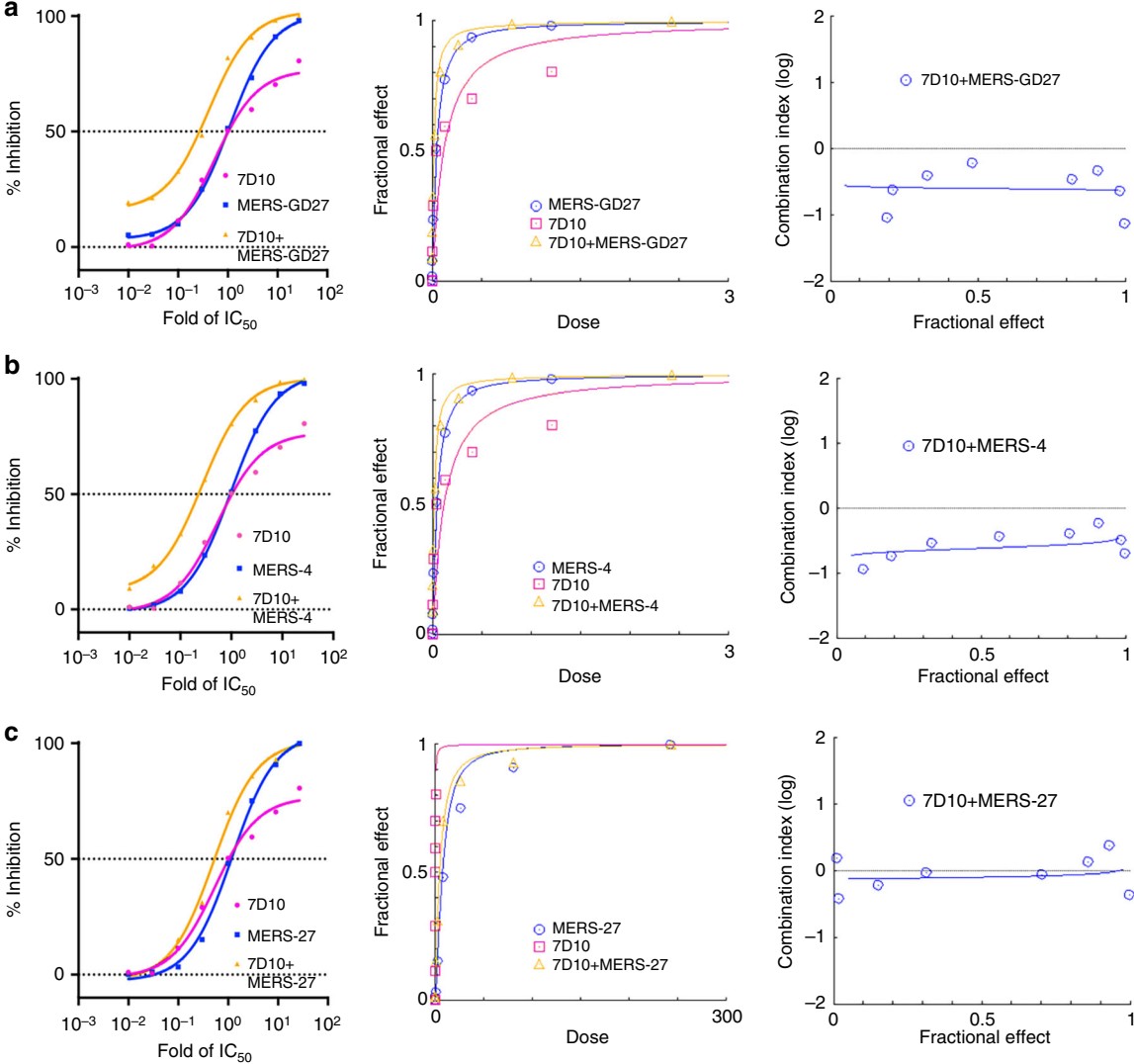

**Fig. 4** Combination effects for 7D10-H with RBD-specific mAbs. **a** Effects of 7D10-H combined with MERS-GD27 in neutralizing pseudotyped MERS-CoV. Percent neutralization was calculated for serial threefold dilutions of each antibody alone and in combination at constant ratios in a range of concentrations from 27 times to 1/81 of $IC_{50}$s. The constant ratios of the combined antibodies were their $IC_{50}$s. On the x-axis, a dose of 1 was at the $IC_{50}$ concentration. Fractional effect (FA) plots generated by the CompuSyn program for 7D10, MERS-4 and their combination showing dosage versus effect. Median effect plot of calculated CI values (logarithmic) versus FA values, in which a log CI of <0 indicates synergism, a log CI of >0 indicates antagonism and a log CI of =0 indicates additive action. The percent neutralization, fractional effect, and CI values for 7D10 combined with MERS-4 (**b**) and 7D10 combined with MERS-27 (**c**) were calculated and generated using the same method. Source data are provided as a Source Data file

and the influence by protease cleavage, receptor binding and antibodies can be biochemically studied by monitoring the appearance of a proteinase-K-resistant band in the sodium dodecyl sulphate (SDS-PAGE) gel comprising the postfusion six-helix bundle[30,31]. We showed that the MERS-CoV S glycoprotein in the prefusion state is sensitive to the digestion of proteinase K (Fig. 6c). Previous studies have demonstrated that cleavage at the S1/S2 site by trypsin and the binding with cellular receptor greatly enhanced the prefusion to postfusion transition of the spike glycoprotein[31]. Consistently, the amount of a 50 kDa and proteinase-K-resistant band of the S glycoprotein representing the postfusion six-helix bundle was at the maximum level in the presence of trypsin and DPP4 (Fig. 6c). And the addition of 7D10-H Fab obviously reduced the intensity of the band (Fig. 6c). Meanwhile, we analyzed the full-length MERS-CoV S trimer embedded in the membrane of pseudotyped virus and the trigger we used to induce the conformational transition was the incubation with Huh 7 cells that endogenously expressing

DPP4 receptor. After incubating the pseudotyped virus with Huh 7 cells for 1 h at 37 °C, a proteinase-K resistant band on the SDS-PAGE gel appeared and the addition of 7D10-H, 7D10-H Fab, or 7D10 scFv all clearly decreased the intensity of this band (Supplementary Fig. 10). Thus, these biochemical results strongly suggest that 7D10 could also exert its neutralizing activity in the postattachment stage after receptor-binding by inhibiting the conformational transition of the S glycoprotein required for membrane fusion (Fig. 6d).

## Discussion

Since the emergence of MERS-CoV in 2012, effective measures countering its infection have become a major research focus. Although no anti-MERS-CoV therapy is available yet, neutralizing mAbs and inhibitory peptides against S glycoprotein have demonstrated efficacy against MERS-CoV infection in cells and animal models[18,32–35]. Most reported neutralizing antibodies target the RBD to block its interaction with the cellular receptor

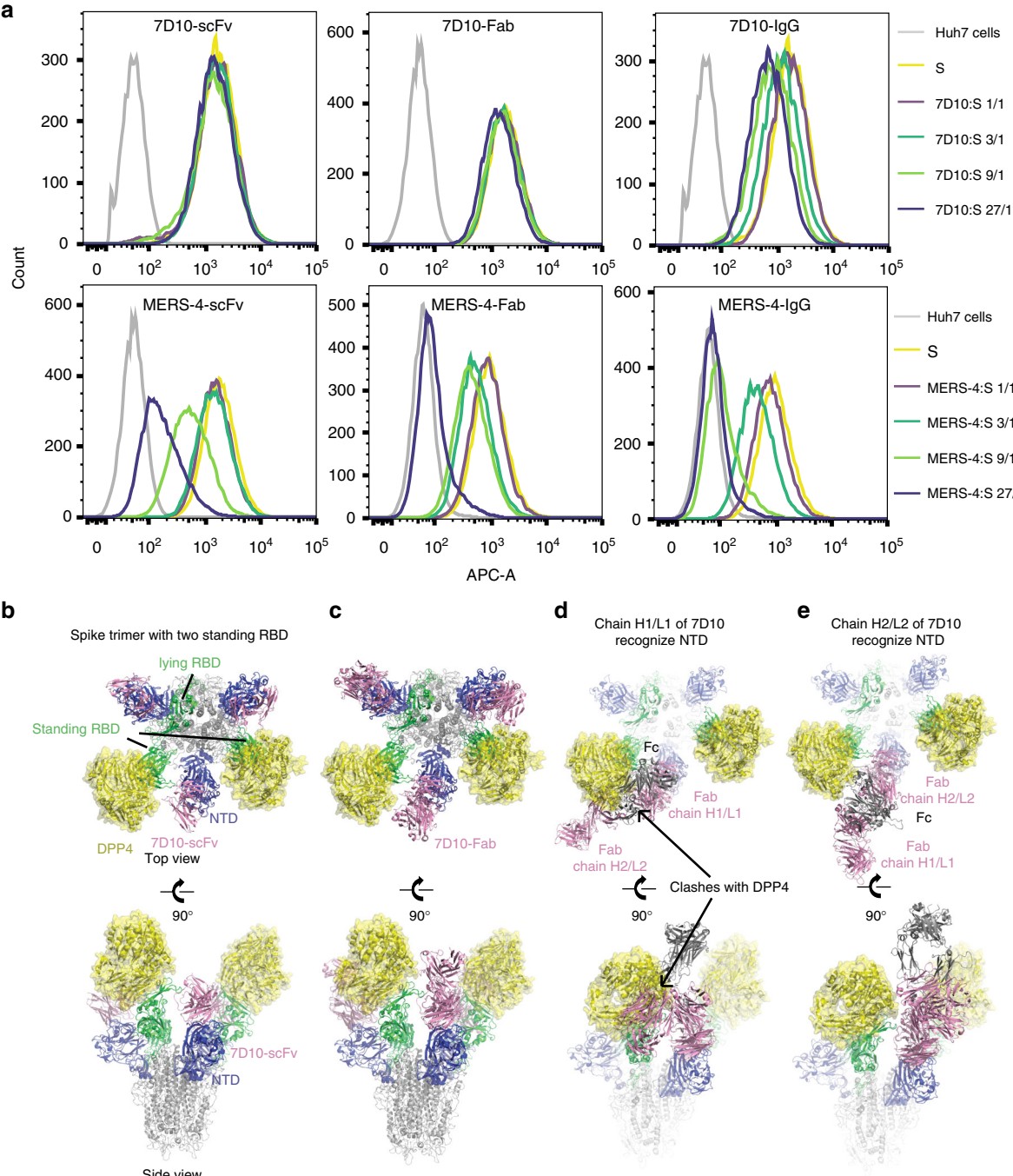

**Fig. 5** 7D10-H partially inhibiting virus binding to its receptor DPP4. **a** Inhibition of the binding between recombinant soluble MERS-CoV spike trimer (S) and human DPP4 expressed on the Huh7 cell-surface. Soluble MERS-CoV spike trimer (S) with strep-tag (1 μg) was incubated with monoclonal antibodies (mAbs) in advance at a molar ratio 1:1, 1:3, 1:9, and 1:27 for 1 h. Huh7 cells were incubated with S or S and mAbs mixtures for 1 h. After washing the unbound S, Huh7 cells were stained with streptavidin APC and analyzed by fluorescence-activated cell sorting (FACS). The amounts of S-bound Huh-7 cells were measured and characterized as median fluorescence intensity. See also Supplementary Table 4 and Supplementary Fig. 6. **b–e** Top view and side view of the MERS-CoV spike trimer in receptor-binding activated states (PDB: 5 × 5c [https://doi.org/10.2210/pdb5X5C/pdb]) with two RBD in the up positions, on which the RBD/DPP4 and NTD/antibody structures are superimposed. The spike trimer (RBD in green, NTD in blue, and S2 subunit in gray) was shown as a cartoon. The DPP4 was shown as a semitransparent surface and colored in yellow. 7D10-scFv (**b**), 7D10-Fab (**c**), and 7D10-IgG (**d**, **e**) were all colored pink. 7D10-Fab and 7D10-IgG were modeled using homologous modeling in SWISS-MODEL

DPP4, which is a critical step for viral cell attachment. In this study, we first isolated the neutralizing mouse antibody 7D10 targeting the NTD of the S glycoprotein. Neutralization assays showed that 7D10 is highly potent and its activity is comparable to that of the most potent RBD-targeting antibodies. Structural determination of 7D10 scFv bound to the NTD and mutagenesis studies revealed the epitope and key residues on the NTD for binding and neutralization at atomic level. Comparisons of 7D10 scFv, Fab, and IgG forms in DPP4-binding competition and neutralization assays indicated that its activity is not solely dependent on the inhibition of DPP4 binding. Further experiments indicated that the neutralizing activity of 7D10 after cell

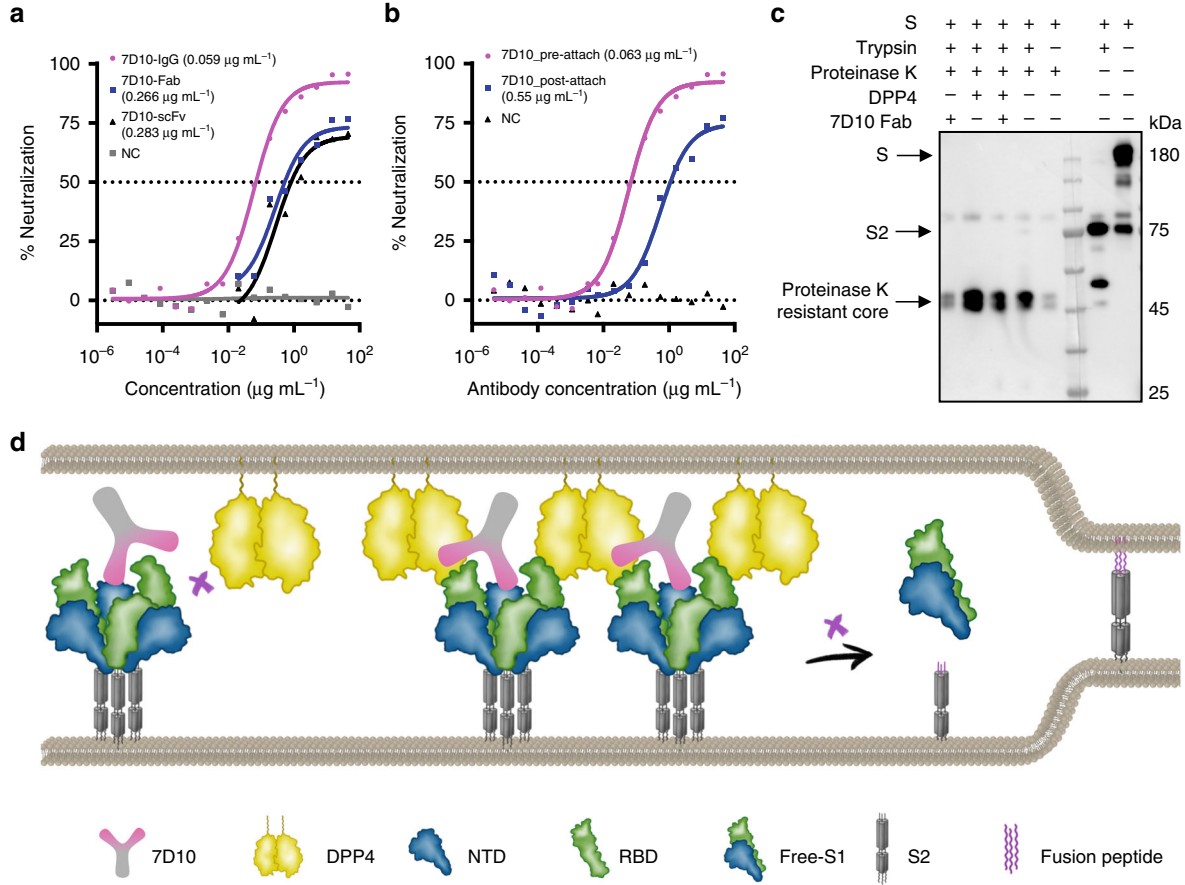

**Fig. 6** 7D10-H retaining neutralizing activity after viral cell attachment. **a** Neutralization curves. 7D10-H IgG, 7D10-H Fab, 7D10 scFv were tested for neutralizing activities against pseudotyped MERS-CoV. VRC01 mAb was used as unrelated control. **b** Pre- and post-attachment neutralizing activities. 7D10-H IgG was tested for neutralizing activity against pseudotyped MERS-CoV before or after receptor binding. VRC01 mAb was used as unrelated control. **c** The effect of 7D10-H Fab on the conformational change of the MERS-CoV S trimer was probed by western blotting using an anti-MERS-CoV S2 polyclonal antibody. Refolding to the postfusion conformation was detected by the appearance of a proteinase-K resistant band. Trypsin was used at 5 μg mL⁻¹ and proteinase K at 10 μg mL⁻¹. Digestion experiments and western blots were performed in triplicates, and a representative result is shown for each of them. **d** A cartoon representation designed by us showing the neutralization mechanism by which 7D10 blocks MERS-CoV entry. On the one hand, some virus particles can not bind to DPP4 due to steric hindrance caused by 7D10 binding. On the other hand, 7D10 still recognizes the particles when the up receptor-binding domain (RBD) binds to DPP4, and may inhibit the prefusion to postfusion transition of the S2 subunit and the initiation of membrane fusion. Source data are provided as a Source Data file

attachment is through the inhibition of prefusion to postfusion conformational transition of the S glycoprotein trimer, which mediates the fusion of viral and cell membranes. We also showed that 7D10 has a wide neutralization breadth against MERS-CoV variants bearing naturally occurring mutations and exhibited synergistic effects with several RBD-targeting antibodies. These results collectively revealed an antibody epitope and neutralization mechanism on the S glycoprotein, which would contribute to the global efforts to control MERS-CoV infection and transmission by providing alternatives for MERS-CoV immunotherapy.

Similar the NTDs of the S protein of other betacoronaviruses such as MHV, BCoV and HKU1, that of MERS-CoV also folds into a galectin-like structure. Although the galectin domain is a typical carbohydrate-recognition domain, the betacoronavirus NTDs can include structural variations that enable more diverse functions in viral infection. The examples, include the NTD of BCoV that retains the glycan-binding activity recognizing 5-N-acetyl-9-O-acetylneuraminic acid (Neu5,9Ac2) and the NTD of MHV that evolved specific protein–protein interactions with its cellular receptor CEACAM1, and both interactions are important for the viral cell attachment[36,37]. However, there is still no report on the glycan or protein-binding activities of the MERS-CoV NTD. In fact,

crystallographic structure determination showed that the glycan-binding site on the MERS-CoV NTD is occupied by a short helix (residues 222–231) and the Asn222-linked glycan, indicating that it is not able to bind glycans in the same way as the NTD of BCoV[11]. Notably, the Asn222-linked glycan is involved in the recognition by 7D10, whereby NAG508 and MAN519 undergoes specific hydrogen-bonding interactions with Tyr58 and Arg98 of 7D10, respectively. The NTD N222Q mutation also dramatically reduced the binding and neutralization by 7D10, but did not dramatically affect the cell infection of pseudotyped MERS-CoV (Supplementary Fig. 11). Therefore, the Asn222-linked glycan serves as an important anchor point for the binding of 7D10 to the MERS-CoV NTD.

As the largest class I viral fusion protein, the coronavirus S glycoprotein is expected to undergo a prefusion to postfusion conformational transition to mediate the interaction between viral and cellular membrane proteins, although structural studies just began to shed light on this recently. The S glycoprotein of beta-coronaviruses MHV and HKU1, whose structures have been determined by the cryo-EM method, all adopt a similar prefusion homotrimeric architecture[38,39]. Interestingly, in the prefusion architecture of the S trimer of highly pathogenic MERS-CoV and SARS-CoV, two major conformational states were observed. A

major difference between them is the change of the RBD in the S1 subunit from a down to an up position, which was proposed to be a prerequisite for the binding of the S trimer to their respective cellular receptor DPP4 and ACE2[9]. This proposal was recently confirmed by our cryo-EM study of the SARS-CoV S trimer in complex with ACE2, and we also showed that ACE2-binding could induce the dissociation of the S1 subunit, which results in the falling apart of the prefusion S trimer and the transition to the prefusion state of the S2 subunit[12]. A major neutralization mechanism of antibodies against MERS-CoV is to directly or indirectly compete with the cellular receptor DPP4 for binding to the RBD. In theory, antibodies that interfere with the coronavirus membrane fusion process other than receptor binding would also have a neutralizing activity, and the 7D10 mAb targeting the NTD we studied is one such example. Here, we showed that 7D10 neutralization is not solely dependent on DPP4-binding competition, and its inhibition of the S trimer conformational transition after cell attachment also plays a significant role in the neutralization. We suggested that the binding of 7D10 may stabilize the prefusion architecture of the S trimer, even after the binding of DPP4 receptor. The stabilization of viral fusion protein at one conformational state for neutralization has also been observed and studied in other viruses such as HIV. A recent study revealed that the HIV Env trimer is intrinsically dynamic with three major and distinct prefusion conformations[40]. Among them, the closed, ground-state conformation is dominant and could be remodeled to another two conformations by CD4 receptor binding, which is essential for the subsequent prefusion to postfusion transition[40]. The binding of neutralizing antibodies, whether inhibiting the binding of the CD4 receptor (such as VRC01) or not (such as 2G12 and PGT145) all resulted in the stabilization of the ground-state conformation of the Env, which finally disfavors its prefusion to postfusion state transition required for viral entry[40,41].

To the best of our knowledge, our study offers the first structural definition of the neutralizing epitope of an antibody targeting the S NTD of MERS-CoV. As we summarized in Supplementary Table 5, a total of six anti-NTD mAbs have been reported[20,21,26,42]. All of them neutralize the infection of pseudotyped MERS-CoV EMC strain with high potency except for mAb 1.10f3. The mAb 5F9 and our 7D10 showed the same neutralizing activity against live MERS-CoV in plaque reduction neutralization testing. Notably, the mouse mAb G2 can greatly relieve the symptom of DPP4-transgenic mice infected following MERS-CoV infection and our 7D10-H can inhibit the infection of pseudotyped MERS-CoV in R26-hDPP4 mice. However, the specific neutralizing epitopes and mechanisms of 5F9, G2, JC57-13, and FIB-H1 are largely unknown. In addition, the combination of different antibodies is supposed to be an effective strategy to combat MERS-CoV infection as it continues to spread among multiple animal species and to probe and adapt to the human population[43–45]. An effective combination would require the candidate antibodies to bind to disparate epitopes or with distinct mechanisms and hence display additive or synergistic effects, as the mAbs MERS-4 and 5F9 we mentioned before[25]. Although the exact mechanism that leads to the synergy or additive is uncertain, our 7D10-H with MERS-GD27 or MERS-4 antibodies demonstrated a synergy in inhibiting the infectivity of pseudotyped MERS-CoV, while 7D10-H and MERS-27 antibodies together had an additive effect. Consequently, 7D10 is currently the most comprehensively studied NTD-targeting mAb with a different epitope and working mechanism, which makes it an excellent candidate, in combination with other RBD-targeting neutralizing antibodies or alone, in our battle against MERS-CoV infection.

## Methods

**Ethics statement**. All studies were performed in compliance with animal protocols (#2017-B-004) approved by the Institutional Animal Care and Use Committee of the National Institute for Food and Drug Control, China Food and Drug Administration (CFDA, Beijing, China) and in compliance with the "Guide for the Care and Use of Laboratory Animals" (National Academies Press: Washington, DC, USA, 2011; 8th ed.). The license number of the Animal Use Certificate issued by the Science & Technology Department of China (Beijing, China) was SYXK 2016-004, approved on 18 February 2016. All experiments associated with live MERS-CoV were conducted in a BSL-3 laboratory at the National Institute of Viral Diseases Control and Prevention, China CDC. The institutional biosafety committee approved all experiments involving live MERS-CoV.

**Cell lines, virus, and animals**. Vero E6, 293T, 293F, and Huh7 cell lines were bought from ATCC and cultured in Dulbecco's Modified Eagle medium (DMEM) supplemented with 10% fetal bovine serum (FBS) and incubated at 37 °C in a humidified atmosphere comprising 5% $CO_2$. The MERS-CoV strain (HCoV-EMC/2012) was kindly provided by Professor Ron Fouchier (Erasmus Medical Centre, Rotterdam, Netherlands). Female BALB/c mice aged 6–8 weeks were used for mAb production. Genetically modified R26-hDPP4 mice aged 4 weeks were used for protection assay. BALB/c mice were purchased from Beijing Vital River Laboratory Animal Technology Co., Ltd. (licensed by Charles River) and housed in specific pathogen free mouse facilities. Genetically modified mice were supplied by the Institute for Laboratory Animal Resources, National Institute for Food and Drug Control (Beijing, China).

**Mouse immunization and mAb generation**. Mice were immunized with 35 µg MERS-CoV S (residue 1–1297) (Sino Biological) combined with 150 µL Freund's complete adjuvant (Sigma, St. Louis, CA, USA) via subcutaneous immunization. Three weeks after the initial immunization, these mice were boosted twice at 2-week intervals. Cells collected from the spleens of sacrificed animals were fused with cultured SP2/0 cells at a 10:1 ratio in the presence of PEG1450 (Sigma). HAT selection medium was used for the fused hybridoma cultures. After 2-weeks of incubation, the positive hybridomas were selected via S-coated ELISA, and the positive clones were subjected to limited dilutions and downstream validation. For large-scale mAb production, ascites fluid from mice inoculated with the hybridomas was collected and purified by the caprylic acid-ammonium sulfate precipitation method.

**Protein expression and purification**. The coding sequence of the MERS-CoV spike glycoprotein ectodomain (EMC strain, spike residues 1–1290) was ligated into the pFastBac-Dual vector (Invitrogen) with a C-terminal T4 fibritin trimerization domain and a hexa-His-strep tap tag to facilitate further purification processes. Briefly, the protein was prepared using the Bac-to-Bac baculovirus expression system, purified by sequentially applying Strep-Tactin and Superose 6 column (GE Healthcare) with HBS buffer (10 mM HEPES, pH 7.2, 150 mM NaCl). Fractions containing MERS-CoV S glycoprotein were pooled and concentrated for subsequent biochemical analyses. The sequence encoding MERS-CoV S1 NTD (residues 18–353) with a C-terminal hexa-His tag was inserted into the eukaryotic expression vector pVAX. FreeStyle 293-F cells were transfected with the plasmid using polyethylenimine (PEI) (Sigma). After 72 h, the supernatant was collected and the NTD was purified using NTA sepharose (GE Heathcare) and Superdex 200 High Performance column (GE Healthcare) with HBS buffer (10 mM HEPES, pH 7.2, 150 mM NaCl).

The sequence encoding the 7D10 $V_L$ and $V_H$ were separately cloned into the backbone of antibody expression vectors containing the constant regions of human IgG1. The chimeric antibody 7D10-H was expressed in FreeStyle 293-F cells by transient transfection and purified by affinity chromatography using Protein A Sepharose and gel-filtration chromatography. The purified 7D10-H was exchanged into phosphate-buffered saline (PBS), and was digested with papain protease (Sigma) over night at 37 °C. The digested antibody was then passed back over Protein A Sepharose to remove the Fc fragment, and the unbound Fab in the flow through was additionally purified using a Superdex 200 High Performance column (GE Healthcare). The gene encoding the 7D10 $V_L$ followed by $V_H$ with a connecting triple GGGS linker and a C-terminal hexa-His tag was synthesized and cloned into the eukaryotic expression vector pVRC8400. FreeStyle 293-F cells were transfected the plasmid in the presence of PEI (Sigma). The cell-culture supernatant was collected 72 h after the transfection, and the 7D10 scFv was collected and captured on NTA Sepharose (GE Healthcare). The bound 7D10 scFv was eluted with HBS buffer containing 500 mM imidazole and was then further purified by gel-filtration chromatography using a Superdex 200 High Performance column (GE Healthcare).

**Complex preparation and crystallization**. The MERS-CoV NTD and the scFv fragment of 7D10 were mixed at a molar ratio of 1:1.2, incubated for 2 h at 4 °C and further purified by gel-filtration chromatography. The purified complex concentrated to approximately 10 mg mL$^{-1}$ in HBS buffer (10 mM HEPES, pH 7.2, 150 mM NaCl) was used for crystallization. The screening trials were performed at 18 °C using the sitting-drop vapor-diffusion method by mixing 0.2 µL of protein with 0.2 µL of reservoir solution. Initial crystallization conditions were obtained in the Crystal Screen Kits (Hampton) and Structure Screen Kits (Molecular Dimensions). The optimized crystals used for diffraction data collection were obtained in a 0.2 M potassium sodium tartrate, 0.1 M sodium citrate pH 5.6, 2.0 M ammonium sulfate, and 10% (v/v) of the additive acetonitrile (40% v/v).

**Data collection and structure determination**. To collect the diffraction data, all crystals were flash-cooled in liquid nitrogen after being incubated in reservoir solution containing 20% (v/v) glycerol. The diffraction images were collected on the BL17U beamline at the Shanghai Synchrotron Research Facility (SSRF)[46] with the wavelength of 0.9796 Å. All images were processed with HKL2000[47]. The structure was solved by molecular replacement using PHASER from the CCP4 suite[48]. The search models were the MERS-CoV NTD structure (PDB ID: 5vyh) and the structures of the variable domain of the heavy and light chains available in the PDB with the highest sequence identities. Subsequent model building and refinement were performed using COOT and PHENIX, respectively[49,50]. There are 94% of most favored, 5.3% of allowed and 0.3% of disallowed Ramachandran plot in the final refinement model. All structural figures were generated using PyMOL[51].

**Neutralizing assay of pseudotyped MERS-CoV**. 293T cells cultured in 100 mm dish were co-transfected with 6 μg of pcDNA3.1-MERS-Spike or its mutants and 24 μg of pNL4-3.luc.RE. The supernatants containing sufficient pseudotyped MERS-CoV were harvested 48–72 h post-transfection. Subsequently, the 50% tissue culture infectious dose (TCID$_{50}$) was determined by infection of Huh7 cells. For the neutralization assay, 100 TCID$_{50}$ per well of pseudoytped virus were incubated with 16 or 8 serial 1:3 dilutions of purified antibodies, Fabs or scFvs for 1 h at 37 °C, after which Huh7 cells (about $1.5 \times 10^4$ per well) were added. After incubation for 72 h at 37 °C, the neutralizing activities of antibodies were determined by the luciferase activity and presented as IC$_{50}$, calculated using the dose-response inhibition function in GraphPad Prism 5 (GraphPad Software Inc.)

**Cell entry of pseudotyped virus**. The concentration of the harvested pseudotyped virions was normalized by p24 ELISA kit (Beijing Quantobio Biotechnology Co., Ltd., China) before infecting the target Huh7 cells. The infected Huh7 cells were lysed at 48 h after infection and viral entry efficiency was quantified by comparing the luciferase activity between pseudotyped viruses bearing the mutant- and wild-type MERS-CoV spike glycoproteins.

**Postattachment neutralization assay**. For the postattachment pseudotyped virus neutralization assay, Huh7 cells, upon reaching a density of $1.5 \times 10^4$ per well in a 96-well plate, were incubated with 100 TCID$_{50}$ per well of pseudotyped virus at 4 °C for 1 h. After removing the supernatant, 200 μL of PBS was added twice to each well to wash the un-bond pseudotyped viruses. A total of 16 serial 1:3 dilutions of purified antibodies in DMEM (10% FBS) were then added to the Huh 7 cells with attached pseudotyped viruses, as well as DMEM (10% FBS) alone as control. Neutralization activities were determined based on the luciferase activity after incubation for 72 h at 37 °C and also presented as IC$_{50}$, calculated using the dose–response inhibition function in GraphPad Prism 5 (GraphPad Software Inc.)

**Cooperativity of mAbs for neutralization**. Synergistic, additive, and antagonistic interaction between 7D10 and MERS-GD27, 7D10, and MERS-27, as well as 7D10 and MERS-4 for virus neutralization were evaluated by the median effect analysis method using CompuSyn software as previously reported[52,53]. The measured neutralization values were input to the program as fractional effects (FA) ranging between 0.01 and 0.99 for each of the two antibodies and for both in combination. CI values were calculated in relation to FA values. A logarithmic CI value of 0 indicates an additive effect, <0 indicates synergism, and >0 indicates antagonism.

**Live MERS-CoV neutralization assay**. The neutralizing activity of the mAbs against live MERS-CoV was also determined in DPP4-expressing Vero E6 cells. Upon reaching a density of $5 \times 10^4$ per well in a 12-well plate, cell monolayers were infected with 30–35 plaque-forming units (PFU) of live virus in the presence or absence of the mAb. After three days of incubation at 37 °C, the inhibitory capacity of the mAbs was assessed by determining the numbers of plaques compared with the potent MERS-CoV anti-RBD and anti-N9 mAbs.

**Murine model of MERS-CoV pseudovirus infection**. The MERS-CoV susceptible animal model hDPP4-knockin mouse, which was established by inserting human dipeptidyl peptidase 4 (hDPP4) into the Rosa26 locus using CRISPR/Cas9, resulting in global expression of the transgene in a genetically stable mouse line[28], was used in this experiment. Mice ($N = 5$) were challenged by intraperitoneal injection (I.P.) with doses of $1.27 \times 10^{7.5}$ TCID$_{50}$ of pseudotyped MERS-CoV. 7D10-H and MERS-4 were administered I.P. to R26-hDPP4 mice at a dose of 200 μg per mouse prior to challenge with pseudovirus. Mice ($N = 4$ for the PBS group and $N = 3$ for the 3C11 group) were also administered PBS or control mAb 3C11 (mAb of anti-NA of H5N1, at a dose of 400 μg per mouse) and challenged using the same I.P. dose of pseudovirus. The IVIS-Lumina II imaging system (Xenogen, Baltimore, MD, USA) was used to detect bioluminescence. Prior to measuring luminescence, the mice were anesthetized using an I.P. injection of sodium pentobarbital (240 mg kg$^{-1}$). The exposure time was 60 s, and fluorescence intensity in regions of interest was analyzed using Living Image software (Caliper Life Sciences, Baltimore, MD, USA). Different wavelengths were used for detecting pseudovirus and tdTomato fluorescence. The substrate, D-luciferin (50 mg kg$^{-1}$, Xenogen-

Caliper Corp., Alameda, CA, USA), was injected I.P. and imaging was conducted 10 min later. The relative intensities of emitted light were represented as colors ranging from red (intense) to blue (weak) and quantitatively presented as photon flux in photons s$^{-1}$ cm$^{-2}$ sr$^{-1}$.

**Binding studies using BLI**. Binding kinetics of MERS-CoV NTD and its mutants with 7D10 were studied using a FortéBio Octet HTX instrument. Assays with agitation set to 1000 rpm in HBS buffer (10 mM HEPES, pH 7.2, 150 mM NaCl) supplemented with 0.01% (v/v) Tween 20 were performed at 25 °C in solid black tilted-bottom 96-well plates (Greiner Bio-One). 7D10 (20 μg mL$^{-1}$) was used to load anti-human IgG Fc capture probes for 300 s to capture levels of 0.5–1 nm. Biosensor tips were then equilibrated for 300 s in HBS buffer supplemented with 0.01% (v/v) Tween 20 prior to binding assessment with different concentrations of wild-type or mutant MERS-CoV NTD for 120 s, followed by dissociation for 240 s. Data analysis and curve fitting were performed using Octet software, version 9.0.

**Binding competition assays by SPR**. Real-time binding and analysis by SPR were conducted on a BIAcore T200 instrument with CM5 chips (GE Healthcare) at room temperature. For all the analyses, HBS buffer consisting of 10 mM HEPES, pH 7.2, 150 mM NaCl and 0.005% (v/v) Tween 20 was used, and all proteins were exchanged to the same buffer. The blank channel of the chip was used as the negative control. DPP4 (20 μg mL$^{-1}$) was immobilized on the chip at about 100 response units. Soluble MERS-CoV spike trimer (S) at the same gradient in the present or absence of the concentration gradient of IgGs, Fabs, or scFvs was flowed over the chip surface. After each cycle, the sensor surface was regenerated with 7.5 mM NaOH. Data were analyzed using the BIAcore T200 evaluation software by fitting to a 1:1 Langmuir binding model.

**FACS analysis of cell-surface staining**. The binding between recombinant soluble MERS-CoV spike trimer (S) and human DPP4 expressed on the surface of Huh7 cells was measured using fluorescence-activated cell sorting (FACS). All cell-surface staining experiments were performed at room temperature. Soluble MERS-CoV spike trimer (S) with strep-tag (1 μg) was incubated with monoclonal antibodies (mAbs) in advance at molar ratios of 1:1, 1:3, 1:9, and 1:27 for 1 h. Huh7 cells were trypsinized and then incubated with S or S and mAbs mixtures for 1 h. After washing the un-bound S with PBS 3 times, the Huh7 cells were then stained with streptavidin APC (BD eBioscience) for another 45 min. Cells were subsequently washed with PBS 5 times and analyzed by flow cytometry on a FACS Aria III machine (BD eBiosciences).

**Western blots**. Totally, 10 μL pseudotyped MERS-CoV was thawed and mixed with 2 μg of antibodies (IgG, Fab or scFv) for 1 h. The virus alone or the mixture was incubated with 20 μL of Huh7 cell suspension for another 1 h at 37 °C. An equal volume of buffer and proteinase-K (final concentration of 10 μg mL$^{-1}$; Thermo_Fisher) was then added and incubated 1 h at 4 °C. For the soluble S, 1 μg of the S trimer was incubated with 3 μg of the DPP4 ectodomain or 3 μg of 7D10 Fab for 1 h on ice. Trypsin (final concentration of 5 μg mL$^{-1}$; Thermo_Fisher) was then added to these samples and incubated 30 min at 37 °C. Subsequently, the samples were supplemented with 10 μg mL$^{-1}$ proteinase-K and incubated 30 min at 4 °C. 6× SDS-PAGE loading buffer was then added to all samples prior to boiling at 100 °C. Samples were run on a 4–12% gradient Tris-MOPS-Gel (GenScript) and transferred to polyvinylidene fluoride membranes. An anti-S2 MERS-CoV S polyclonal antibody (1:2000 dilution; Thermo_Fisher; Cat#PA5-81788) and an HRP-conjugated goat anti-rabbit secondary antibody (1:500 dilution; HuaxingBio; Cat#HX2027) were used for Western blotting. AI600 was used to develop images.

**Reporting summary**. Further information on research design is available in the Nature Research Reporting Summary linked to this article.

## Data availability
The source data underlying Figs. 1A–D, 3, 4, and 6A–C and Supplementary Figs. 1A–C, 3, 4, 7, 9–11 are provided as a Source Data file. Crystal structures presented in this work has been deposited in the Protein Data Bank (PDB) and are available with accession code 6J11.

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

## Acknowledgements

We would like thank Dr. Changfa Fan (Division of Animal Model Research, Institute for Laboratory Animal Resources, National Institutes for Food and Drug Control) for help in providing the R26-hDPP4 mouse model and experimental method. We thank Dr. Jianhua He and the staff scientists at the SSRF BL17U beam line, as well as Dr. Shilong Fan at the X-ray crystallography platform of the Tsinghua University Technology Center for assistance in diffraction data collection. This work was supported by the National Key Plan for Scientific Research and Development of China (grants 2016YFD0500307 and 2016YFD0500301), the National Natural Science Foundation of China (grants 31470751 and U1405228), and the National Major Project for Control and Prevention of Infectious Disease in China (2016ZX10004001-003).

## Author contributions

H.Z., W.T., L.Z. and X.W. designed the experiments. Y.C., K.Q. and W.T. isolated the antibody 7D10 and sequenced the corresponding $V_L$ and $V_H$ genes. B.H. carried out the neutralizing assay with live MERS-CoV. H.Z. and S.Z. expressed, purified, and crystallized the protein, and H.Z. carried out the BLI and SPR analysis. H.Z. conducted all the neutralizing assays based on pseudotyped MERS-CoV with the help of W.J. H.Z. conducted DPP4-competition assays and the Western Blots analysis. P.N. performed the protection assay in mice. H.Z. and J.L. collected the diffraction data. H.Z. and X.W. processed the diffraction data, determined, and analyzed the structure. H.Z. and X.W. wrote the paper with contributions from L.Z. and W.T.

## Additional information

**Competing interests:** The authors declare no competing interests.

