## [Peer Review File · Nature Communications]

Point-by-Point response to reviewers

Reviewers' comments:

Reviewer #1 (Remarks to the Author):

Zhou et al. present impressive work describing a new mouse antibody (7D10) that binds to the N-terminal domain of MERS spike protein and is capable of neutralizing several variants of MERS-CoV. The authors performed the experiments with a chimeric version of the antibody containing the human IgG1 backbone and elaborate upon the initial mAb characterization by determining the antibody epitope with crystallography and extensive mutagenesis. The crystal structure of the complex reveals interesting interactions between the antibody and NTD Asn222-liked glycans. This work describes in detail the novel epitope and neutralization breadth of a newly isolated mouse antibody that binds to the S1 N-terminal domain rather than the more characterized RBD domain.

Major points:

1. Although the authors are the first to describe the molecular interactions of an antibody to S1-NTD, there are other NTD antibodies that have been characterized over the recent years. There is no reference or comparison to other MERS-NTD antibodies like G2 (PMID: 26218507), 5F9 (PMID: 28536429), JC57-13 and FIB-H1 (PMID: 29514901) in the manuscript.
2. There are discrepancies in the pseudovirus neutralization data presented in Figure 1 and 4. In figure 1C, 7D10 is shown to neutralize 100% of the virus while it only exhibits ~70% neutralization in figures 4A, B and C. The baselines for 0% and 50% neutralization also seem to vary significantly between figures 4A, B and C. The authors should explain how the normalization was performed for these datasets. Notably, other NTD Abs (PMID: 29514901) have been shown to have low IC50 values but much higher or no IC90 values at relevant concentrations.
3. Despite the author's statement in the abstract that they reveal the neutralization mechanism, the mechanism of 7D10 neutralization is still undefined. The pre- and post-attachment studies as well as the increased potency of IgG compared to Fab and scFv indicate that 7D10 moderately blocks RBD-DPP4 interactions by steric hindrance but the key mechanism of 7D10 activity is still unknown. In the discussion, the authors hypothesize that 7D10 might neutralize by stabilizing the spike in its pre-fusion conformation. Any preliminary experimental evidence supporting this hypothesis should be included in the manuscript.

Minor points:

1. For clarity, authors should use the same color for 7D10 in all binding and neutralization graphs presented.
2. Authors present data demonstrating synergistic activity for 7D10 with other RBD antibodies in neutralizing MERS-CoV. Synergism between spike NTD and RBD antibodies have been described before and should be referenced (PMID: 29514901).
3. The authors also comment on 7D10 being a very promising candidate to battle against MERS-CoV. It would be interesting to see if 7D10 protects against MERS-CoV challenge in animal models if given prophylactically or therapeutically.

Reviewer #2 (Remarks to the Author):

The manuscript "Structural definition of a new neutralization epitope on the N-terminal domain of MERS-CoV spike glycoprotein" by Haixia Zhou et al isolated a new a monoclonal antibody 7D10 from immunized mice that neutralized cell infection of MERS-CoV. Structural and mutagenesis studies revealed that this new antibody binds to a new epitope on the N-terminal domain of the spike glycoprotein that has not been targeted by MERS-CoV antibodies isolated previously. The authors also performed studies that showed 7D10 is synergistic in neutralization with the DPP4-binding site targeting antibodies. Even though the mechanism of 7D10 neutralization remain to be fully elucidated, the characterization and structural studies of this novel antibody is worth publication in Nature Communications.

The paper is well written, and I only have some minor comments:

1. Supplementary tables 2 and 3 were not called out in the manuscript;
2. Page 4, line 12, "confirmation" should be "conformational";
3. It would be nice to show in a supplementary table the hydrogen bonds and salt bridges between 7D10 and NTD.

Reviewers' comments:

Reviewer #1 (Remarks to the Author):

Zhou et al. present impressive work describing a new mouse antibody (7D10) that binds to the N-terminal domain of MERS spike protein and is capable of neutralizing several variants of MERS-CoV. The authors performed the experiments with a chimeric version of the antibody containing the human IgG1 backbone and elaborate upon the initial mAb characterization by determining the antibody epitope with crystallography and extensive mutagenesis. The crystal structure of the complex reveals interesting interactions between the antibody and NTD Asn222-liked glycans. This work describes in detail the novel epitope and neutralization breadth of a newly isolated mouse antibody that binds to the S1 N-terminal domain rather than the more characterized RBD domain.

Response: We appreciate the reviewer's comments. In particular, the suggestion to show the evidence of our hypothesis that 7D10 can stabilize the spike is crucial for better understanding the mechanism of 7D10 action. To this end, we have conducted additional experiments to address the reviewer's questions. Detailed descriptions are listed below.

Below are specific points that should be addressed.

Major points:

1. Although the authors are the first to describe the molecular interactions of an antibody to S1-NTD, there are other NTD antibodies that have been characterized over the recent years. There is no reference or comparison to other MERS-NTD antibodies like G2 (PMID: 26218507), 5F9 (PMID: 28536429), JC57-13 and FIB-H1 (PMID: 29514901) in the manuscript.

Response: We agree with the reviewer that the comparisons of 7D10 with other reported NTD-targeting mAbs are necessary. We added relevant summary in the **Discussion** and **Table 2** in particular on those antibodies' (7D10, 5F9, G2, C57-13, and FIB-H1) neutralization activities against pseudotyped and live MERS-CoV, *in vivo* protection and epitope identification. As shown in **Table 2**, all of them neutralize the infection of pseudotyped MERS-CoV EMC strain with high potency. The mAbs 5F9 and 7D10 have the same neutralizing activity against live MERS-CoV in plaque reduction neutralization testing. Notably, the mouse mAb G2 can relieve the symptoms of DPP4-transgenic mice infected by MERS-CoV and our 7D10-H can inhibit the infection of pseudotyped MERS-CoV in R26-hDPP4 mice. However, the neutralizing epitopes and mechanisms of 5F9, G2, JC57-13 and FIB-H1 are largely unknown.

Table 2. Summary of MERS-CoV S NTD-targeting neutralizing mAbs

mAbs	Species	Pseudotyped MERS-CoV (EMC) neutralization	Live MERS-CoV (EMC) neutralization	In vivo protection	Crystal structures available
7D10	Mouse	The IC ₅₀ was 0.18 µg/ml and the IC ₅₀ of 7D10-H was 0.06 µg/ml.	The PRNT IC ₅₀ was 0.2 µg/ml.	7D10-H can inhibit the infection of pseudotyped MERS-CoV in R26-hDPP4 mice.	Yes, NTD/7D10 scFv
5F9	Mouse	The IC ₅₀ was 0.24 µg/ml.	The PRNT IC ₅₀ was 0.2 µg/ml.	Not reported	Not reported
G2	Mouse	The IC ₅₀ was 0.010 µg/ml	Not reported	G2 can protect against MERS-CoV infection in DPP4-transgenic mice.	Not reported
JC57-13	Rhesus macaques	The IC ₅₀ was 0.0085 µg/ml (also showed as 0.068 µg/ml).	The PRNT IC ₅₀ was not available (N/A), but 50% neutralization was obtained at the concentration of 0.0032 µg/ml.	Not reported	Not reported
FIB-H1	Rhesus macaques	The IC ₅₀ was 0.0083 µg/ml	Not reported	Not reported	Not reported

PRNT: plaque reduction neutralization testing

2. There are discrepancies in the pseudovirus neutralization data presented in Figure 1 and 4. In figure 1C, 7D10 is shown to neutralize 100% of the virus while it only exhibits ~70% neutralization in figures 4A, B and C. The baselines for 0% and 50% neutralization also seem to vary significantly between figures 4A, B and C. The authors should explain how the normalization was performed for these datasets. Notably, other NTD Abs (PMID: 29514901) have been shown to have low IC₅₀ values but much higher or no IC₉₀ values at relevant concentrations.

Response: As suggested, we re-conducted the neutralizing assays of 7D10, 7D10-H, RBD-specific mAbs and the synergistic assays in the same condition. As shown in the following figure, 7D10 neutralized 100% of the virus and 7D10-H can neutralize over 90%, which are consistent with our neutralizing assays before. In Figure 4, the synergistic experiments have three steps: 1) to calculate IC₅₀ of each antibody in neutralizing pseudotyped MERS-CoV, 2) to calculate the inhibition of each antibody alone and in combination in a range of concentrations from 27 times to 1/81 of IC₅₀s, 3) to generate fractional effect and CI values by the CompuSyn program. As shown in the following figure, the IC₅₀ values of 7D10-H, MERS-GD27, MERS-4 and MERS-27 are 0.045 µg/ml, 0.004 µg/ml, 0.04 µg/ml and 9.2 µg/ml, respectively. Thus, in the step 2 (as shown in the left figures of 4A, B and C), the highest concentrations of these antibodies we used are 27 times IC₅₀ (namely 1.215 µg/ml, 0.108 µg/ml, 1.08 µg/ml and 248 µg/ml, respectively). While the highest concentrations of

antibodies used in the step 1 and other neutralizing assays are 48.89 $\mu\text{g/ml}$ or 44.40 $\mu\text{g/ml}$. As shown in the following figure, 7D10-H, MERS-GD27, MERS-4 and MERS-27 can neutralize about 80%, 100%, 100% and 100% respectively of the virus at their 27 times IC_{50} 's concentrations, which are consistent with data showed in Figure 4. The reviewer is right that there were variations of baselines between 4A, B and C before. In fact, the details of each time experiment are variable but the trend of neutralization for a certain antibody is the same. In the initial submission version, the data of 7D10-H and MERS-27 synergistic experiment was not conducted in the same time with others. We agree that we should present the consistency between A, B and C in Figure 4 so we performed the experiment again. And the results of effects between 7D10-H and MERS-GD27, 7D10-H and MERS-4, 7D10-H and MERS-27 still agree with before. The normalization for inhibition is the same at all neutralizing assays, measured by the ratio of the mean difference between experimental holes and cell control to the mean of the virus control. Each set contains three replicates. The IC_{90} of G2 is 0.06 $\mu\text{g/ml}$ and JC57-13 is over 10 $\mu\text{g/ml}$ and there is no IC_{90} value of FIB-H1 (PMID: 29514901). As for our 7D10 and 7D10-H, the IC_{90} s are 2.22 $\mu\text{g/ml}$ and 1.94 $\mu\text{g/ml}$, respectively.

Figure 4 | Combination effects in neutralizing pseudotyped MERS-CoV for 7D10-H with RBD-specific mAbs. (A) Effects of 7D10-H combined with MERS-GD27 in neutralizing pseudotyped MERS-CoV. Percent neutralization was calculated for serial 3-fold dilutions of each antibody alone and in combination at constant ratios in a range of concentrations from 27 times to 1/81 of IC_{50} s. The constant ratios of the combined antibodies were their IC_{50} s. On the x axis, a dose of 1 was at the IC_{50} concentration. Fractional effect (FA) plots generated by the CompuSyn program for 7D10, MERS-4 and their combination showing dosage versus effect. Median effect plot of calculated CI values (logarithmic) versus FA values, in which a log CI of <0 indicates synergism, a log CI of >0 indicates antagonism and a log CI of $=0$ indicates additive action. The percent neutralization, fractional effect, and CI values for 7D10 combined with MERS-4 (B) and 7D10 combined with MERS-27 (C) were calculated and generated using the same method.

3. Despite the author's statement in the abstract that they reveal the neutralization mechanism, the mechanism of 7D10 neutralization is

still undefined. The pre- and post-attachment studies as well as the increased potency of IgG compared to Fab and scFv indicate that 7D10 moderately blocks RBD-DPP4 interactions by steric hindrance but the key mechanism of 7D10 activity is still unknown. In the discussion, the authors hypothesize that 7D10 might neutralize by stabilizing the spike in its pre-fusion conformation. Any preliminary experimental evidence supporting this hypothesis should be included in the manuscript.

Response: This is an excellent suggestion. The retaining activity 7D10-H after viral attachment indicated that 7D10 would also interfere with the pre-fusion to post-fusion conformational transition of the S glycoprotein. It is usually challenging to get a whole structural picture of this dynamic transition procedure. However, this transition and the influence by protease cleavage, receptor binding and antibodies can still be biochemically studied by monitoring the appearance of a proteinase-K-resistant band in the SDS-PAGE gel comprising the post-fusion 6-helix bundle. Thus, we performed the assay to investigate whether 7D10-H inhibits this transition or not. Previous studies have demonstrated that cleavage at the S1/S2 site by trypsin and the binding with cellular receptor greatly enhanced the pre-fusion to post-fusion transition of the spike glycoprotein (PMID: 30712865). Our results showed the MERS-CoV S glycoprotein in the pre-fusion state is sensitive to the digestion of proteinase K, while the amount of a 50 kDa and proteinase-K-resistant band of the S glycoprotein representing the post-fusion 6-helix bundle was at the maximum level in the presence of trypsin and DPP4 (Fig. 6C). And the addition of 7D10-H Fab obviously reduced the intensity of the band (Fig. 6C). Meanwhile, we analyzed the full-length MERS-CoV S trimer embedded in the membrane of pseudotyped virus and the trigger we used to induce the conformational transition is the incubation with Huh 7 cells that endogenously expressing DPP4 receptor. After incubating the pseudotyped virus with Huh 7 cells for 1 h at 37°C, although unlike the soluble S trimer triggered by soluble DPP4, a proteinase-K resistant band on the SDS-PAGE gel still appeared and the addition of 7D10-H, 7D10-H Fab or 7D10 scFv all clearly decreased the intensity of this band (Supplementary Fig. 8). Thus, these biochemical results strongly suggest that 7D10 could also exert its neutralizing activity in the post-attachment stage after receptor-binding by inhibiting the conformational transition of the S glycoprotein required for membrane fusion.

Figure 6 | 7D10-H retaining neutralizing activity after viral cell attachment. (C) The effect of 7D10-H Fab on the conformational change of the MERS-CoV S trimer was probed by western blotting using an anti-MERS-CoV S2 polyclonal antibody. Refolding to the post-fusion conformation was detected by the appearance of a proteinase-K resistant band. Trypsin was used at 5 µg/ml and proteinase K at 10 µg/ml. Digestion experiments and western blots were performed in triplicates, and a representative result is shown for each of them.

Supplementary Fig. 8 7D10-H inhibiting the transition to the post-fusion state of MERS-CoV S visualized by Western Blots. The effect of different forms of 7D10-H on the conformational change of the membrane-embedded S trimer on the MERS-CoV pseudovirus was probed by Western Blots using an anti-MERS-CoV S2 polyclonal antibody. Refolding to the post-fusion conformation was detected by the appearance of a proteinase-K resistant band. Proteinase K was used at 20 $\mu\text{g/ml}$. VRC01 was used as unrelated control. Digestion experiments and western blots were performed in triplicates, and a representative result is shown for each of them.

Minor points:

- 1. For clarity, authors should use the same color for 7D10 in all binding and neutralization graphs presented.**

Response: We modified the Figures with same color for 7D10 in all binding and neutralization graphs in the revised manuscript.

- 2. Authors present data demonstrating synergistic activity for 7D10 with other RBD antibodies in neutralizing MERS-CoV. Synergism between spike NTD and RBD antibodies have been described before and should be referenced (PMID: 29514901).**

Response: The reviewer is correct that synergism between NTD-specific mAb 5F9 and RBD-targeting MERS-4 has been reported in our previous study of MERS-4. We have referenced it in the **Results** and **Discussion** of revised manuscript.

- 3. The authors also comment on 7D10 being a very promising candidate to battle against MERS-CoV. It would be interesting to see if 7D10 protects against MERS-CoV challenge in animal models if given prophylactically or therapeutically.**

Response: We agree with the reviewer that the results of 7D10 in vivo efficacy is very important. We selected the R26-hDPP4 mouse as MERS-CoV susceptible animal model, which was established by inserting human dipeptidyl peptidase 4 (hDPP4) into the Rosa26 locus using CRISPR/Cas9, resulting in global expression of the transgene in a genetically stable mouse line (PMID:30142928). It was shown that high-titer MERS-CoV pseudovirus could also productively infect R26-hDPP4 mice, with effects comparable to the authentic infection. Thus, based on our experiment conditions, we assessed the in vivo efficacy of 7D10-H to protect R26-hDPP4 mice against the infection of pseudotyped MERS-CoV. 7D10-H as well as the RBD-specific mAb MERS-4 at a dose of 200 µg/mouse ablated pseudovirus infection when administered by the intraperitoneal injection (I.P.) (Supplementary Fig. 1D). While the pseudovirus reporter signals in mice administered either PBS or control mAb (3C11) were shown at significantly higher level in the whole body (Supplementary Fig. 1D).

Supplementary Fig. 1 7D10-H sustaining biochemical characters and neutralizing activity. (D) Inhibition of pseudotyped MERS-CoV infection in R26-hDPP4 mice by the mAb 7D10-H. For evaluation of 7D10-H or MERS4, mice (N=5) were administered 200 µg/mouse of mAb I.P. For the unrelated control mAb 3C11, mice (N=3) were administered 400 µg/mouse of 3C11 I.P.

and mice (N=4) were administered PBS as the negative control. And 6 h later with mAbs or PBS, all mice were challenged with pseudovirus I.P. at a dose of $1.27 \times 10^{7.5}$ TCID₅₀. On day 4 and 7 p.i., Bioluminescence imaging of the whole body was conducted and typical images are shown.

Reviewer #2 (Remarks to the Author):

The manuscript “Structural definition of a new neutralization epitope on the N-terminal domain of MERS-CoV spike glycoprotein” by Haixia Zhou et al isolated a new a monoclonal antibody 7D10 from immunized mice that neutralized cell infection of MERS-CoV. Structural and mutagenesis studies revealed that this new antibody binds to a new epitope on the N-terminal domain of the spike glycoprotein that has not been targeted by MERS-CoV antibodies isolated previously. The authors also performed studies that showed 7D10 is synergistic in neutralization with the DPP4-binding site targeting antibodies. Even though the mechanism of 7D10 neutralization remain to be fully elucidated, the characterization and structural studies of this novel antibody is worth publication in Nature Communications.

The paper is well written, and I only have some minor comments:

Response: We appreciate supportive and constructive comments from this reviewer. As suggested, we added a table showing the hydrogen bonds and salt bridges between 7D10 and NTD and corrected the writing errors.

1. Supplementary tables 2 and 3 were not called out in the manuscript;

Response: We have called out all the Supplementary Tables and Figures in the revised manuscript.

2. Page 4, line 12, “confirmation” should be “conformational”;

Response: We have corrected it.

3. It would be nice to show in a supplementary table the hydrogen bonds and salt bridges between 7D10 and NTD.

Response: As suggested, we have added the hydrogen bonds and salt bridges between 7D10 and NTD in Supplementary Table 3.

Reviewer #1 (Remarks to the Author):

The authors have addressed all concerns and the manuscript is now suitable for publication.

Reviewer #2 (Remarks to the Author):

I have read the revised manuscript and am glad that the authors performed additional experiments to provide evidence on the mechanism of 7D10. They also investigated the protective efficacy of 7D10-H against infection of pseudotyped MERS-CoV in R26-hDPP4 mice. Structural details of antibody-antigen complex are now provided in supplementary table 3. The revised manuscript is well written. I have no further comments.